# Determination of the Bentonite Content in Molding Sands Using AI-Enhanced Electrical Impedance Spectroscopy

**DOI:** 10.3390/s24248111

**Published:** 2024-12-19

**Authors:** Xiaohu Ma, Alice Fischerauer, Sebastian Haacke, Gerhard Fischerauer

**Affiliations:** 1Faculty of Engineering Science, University of Bayreuth, 95440 Bayreuth, Germany; xiaohu.ma@uni-bayreuth.de (X.M.); alice.fischerauer@uni-bayreuth.de (A.F.); 2Sensor Control GmbH, 56566 Neuwied, Germany; shaacke@sensor-control.de

**Keywords:** bentonite, quartz sand, electrical impedance spectroscopy (EIS), feature extraction, feature importance, principal component analysis (PCA), foundry, machine learning, fully connected neural networks (FCNN)

## Abstract

Molding sand mixtures in the foundry industry are typically composed of fresh and reclaimed sands, water, and additives such as bentonite. Optimizing the control of these mixtures and the recycling of used sand after casting requires an efficient in-line monitoring method, which is currently unavailable. This study explores the potential of an AI-enhanced electrical impedance spectroscopy (EIS) system as a solution. To establish a fundamental dataset, we characterized various sand mixtures containing quartz sand, bentonite, and deionized water using EIS in the frequency range from 20 Hz to 1 MHz under laboratory conditions and also measured the water content and density of samples. Principal component analysis was applied to the EIS data to extract relevant features as input data for machine learning models. These features, combined with water content and density, were used to train regression models based on fully connected neural networks to estimate the bentonite content in the mixtures. This led to a high prediction accuracy (*R*^2^ = 0.94). These results demonstrate that AI-enhanced EIS has promising potential for the in-line monitoring of bulk material in the foundry industry, paving the way for optimized process control and efficient sand recycling.

## 1. Introduction

In the foundry industry, maintaining the suitable composition of molding sand is crucial to producing high-quality castings. A key component of these sand mixtures is bentonite, a clay additive that enhances the binding properties of sand [1]. Maintaining the active bentonite content within an optimal range is challenging but essential. Too little bentonite can lead to weak molds, resulting in casting defects and structural flaws. Conversely, excessive bentonite can increase costs and can lead to insufficient permeability of the molding sand.

Therefore, precise control over bentonite content is vital to balancing cost-efficiency and product integrity. This need for accuracy underscores the demand for a reliable in-line monitoring system that can assess bentonite levels in real time. The standard method of bentonite determination in molding sand in foundry plants is based on the adsorbed amount of methylene blue [2]. It is a highly labor-intensive method that requires a significant amount of time and specialized technicians to perform the determination. A spectrophotometric method, in conjunction with an infrared spectroscopic approach, was tested in [3] for its potential to assess the quality of active bentonite in molding sand. However, this method still requires considerable manual labor and is not suitable for harsh working environments, such as those exposed to dust and vibration. Electrical conductivity is also used to estimate the bentonite content in soil–bentonite mixtures [4], but careful calibration and adjustments are needed to ensure accurate bentonite content estimation. Additional steps are required to account for varying mixture conditions. This method has provided an initial indication of the feasibility of using electrical techniques for bentonite content estimation.

Studies have further explored the potential of electrical methods for this purpose. Research on the dielectric behavior of bentonite under moist conditions has shown that the properties of bentonite can be studied by recording sample responses to alternating electric fields [5]. Additionally, electrical impedance spectroscopy (EIS) has been effectively applied to monitor the hydration process of cement, demonstrating the versatility of electrical measurements in assessing materials with bentonite inside [6]. Together, these findings indicate that electrical methods, including resistance, impedance, and other related parameters, can serve as effective tools for monitoring bentonite content in foundry mixtures.

Despite the promise of these techniques, a reliable, low-cost, and in-line monitoring system for real-time bentonite content measurement is not yet available and is much needed [7]. To address this need, we propose an in-line monitoring system that combines EIS with machine learning models, specifically fully connected neural networks (FCNNs). EIS measures impedance across a broad frequency range by analyzing the current response to a voltage perturbation, or vice versa, which is known to provide rich information about material properties, interfacial phenomena, and electrochemical reactions [8,9]. It offers several compelling advantages, making it an ideal choice for the in situ monitoring of industrial processes. One of the key benefits is its non-invasive nature, allowing for real-time, continuous analysis without disturbing the system being observed. EIS offers remarkable flexibility in both measurement duration and the volume of material analyzed, rendering it highly adaptable to diverse industrial applications. Specifically, in the context of foundry sand preparation, where the interval between charges typically spans several minutes, varying depending on the specific foundry operations, EIS facilitates the rapid acquisition of critical information within the foundry sands. This efficiency highlights its suitability for dynamic industrial processes requiring timely and precise material characterization. Its ability to capture both the real and imaginary components of impedance over a broad frequency range provides a more comprehensive understanding of material characteristics and system dynamics [8]. This capability, combined with its adaptability, positions EIS as an ideal tool for in situ monitoring in dynamic industrial environments.

The integration of machine learning, particularly AI-driven techniques such as neural networks, enhances the capabilities of EIS by enabling more advanced analysis and interpretation of data [10,11]. This combination is widely used in different areas, including the creation of EIS circuit models and the prediction of circuit parameters [12,13], the estimation of a battery’s state of charge [14,15], and glucose detection for public health and safety in biosensing [16]. AI algorithms combined with EIS represents a powerful approach in materials analysis and industrial monitoring. It simplifies operational complexity by eliminating frequent calibration for variations in system conditions, materials, and environmental factors [17,18]. With sufficient training data derived from EIS measurements and appropriate feature engineering, the impact of these influences can either be minimized or effectively incorporated as input variables in the AI system [19,20,21]. Additionally, combining AI algorithms with EIS enhances the system’s robustness and adaptability in real-time monitoring [22,23].

This study explores the use of neural networks in combination with EIS to accurately determine the bentonite content in foundry sands. Bentonite, a composite material primarily composed of montmorillonite along with other minerals, is known for its high water absorption capacity and low hydraulic conductivity [24,25]. The molding material is typically a mixture of recycled and fresh sand, water, and bentonite with water content around 2% to 12% [26]. Thus, the presence of bentonite can influence the dielectric permittivity of the molding material significantly [27], making EIS an effective method for assessing bentonite content.

In this study, comprehensive impedance measurements of synthetic foundry materials were carefully carried out under laboratory conditions. Relevant features were extracted from the measured impedance data, as raw impedance data obtained from frequency sweeps tend to be high-dimensional and cumbersome for analysis. Alongside extracted impedance features, the physical properties of the mixtures, such as water content and density, were incorporated as input data in FCNNs. Regression FCNNs with various combinations of feature sets were then trained to estimate the bentonite content in foundry sands. Following this, the feature influence was evaluated to identify and eliminate unnecessary features, thereby reducing the complexity and effort required for future measurements.

## 2. Materials and Methods

To evaluate the performance of the AI-enhanced EIS approach under controlled conditions, materials under test (MUTs) were prepared in the laboratory. (The alternative of working with industrial foundry process samples would have led to severe challenges. They are mainly composed of used and fresh sand, potentially active bentonite, and water in unknown and—at least for the water content—time-variant composition. It would have required costly and time-intensive laboratory methods to determine the composition, but even then, the preferred way to characterize a novel measurement method is to apply it to known measurands. This is why we prepared samples under well-controlled conditions.) A total of 100 g of quartz sand with the corresponding amount of sodium bentonite were first mixed in their dry states at different gravimetric ratios, from 1% to 15%. The dry mixtures were then wetted with deionized water at varying levels in transparent polyethylene (LDPE) resealable plastic bags (150 mm × 240 mm), with water content ranging from 1.5 to 10 mass-%. The wet mixtures were thoroughly kneaded within the bags to ensure uniform distribution and left to stand for 10 min, allowing the components to reach a homogeneous state. A total of 230 samples were prepared for subsequent impedance measurements. It is important to note that the water content was calculated as the ratio of the weight of the added water to the weight of the dry mixture. The gravimetric method using oven-drying at 105 °C to determine the water content [28] was not used because of its inherent uncertainties. Our findings indicate that the oven-drying method tends to overestimate the water content, as illustrated in Figure 1.

To shed light on this observation, a Heraeus Model T6060 oven, made by Heraeus Instruments GmbH, Hanau, Germany, was used to dry the materials for 24 h to measure the water content in the mixtures. Mixtures with six bentonite content levels (ranging from 2 g/100 g to 12 g/100 g) and one water content level (4 g/100 g) were investigated. At each bentonite level, five measurements were performed and the average deviation, along with the error bar represented as standard deviation, were plotted in Figure 1. The results indicate that the oven-drying method consistently overestimates water content in foundry mixtures, with the deviation increasing as the bentonite content rises. At lower bentonite content levels (e.g., 2–4 g/100 g), the average and standard deviation are relatively small, showing that measurements are closer to the actual values with less variability. As bentonite content rises (e.g., 8–12 g/100 g), both the average and standard deviation grow. Further analysis revealed that drying bentonite alone in the oven still results in a measurable mass loss. This is likely caused by the fine particle size of bentonite, which becomes airborne and escapes through the oven’s ventilation system during drying [29], leading to an overestimation of water content in the mixtures. Therefore, as mentioned, we determined the water content by weighing the dry mixtures using a mass scale (KERN-PCB-1000-2 with 0.01 g precision) and measuring the weight of the added water.

The electrical properties of the various mixtures were analyzed using the measurement cell depicted in Figure 2. The device consists of two opposing electrodes embedded within the inner wall of a cylindrical chamber with a diameter of 5 cm. The electrodes, made from C45 steel, are arc-shaped, each spanning 120° of the circular inner cylinder with a height of 2 cm. They are insulated from the chamber walls using retaining rings fabricated from polyetheretherketone (PEEK). The electrodes are connected to an Agilent E4980A LCR meter via coaxial cables, enabling precise impedance measurements. The LCR meter is further linked to a computer via a USB connection, allowing the measured impedance spectra to be transferred for processing and analysis. The material under test (MUT) is placed inside the cylinder, where it can be compacted using a manual compressor to ensure firm contact, eliminating any large gaps between the materials.

All the prepared 230 samples were individually filled into the cylinder measuring cell and compressed at random, resulting in a volume range from 60 mL to 90 mL (corresponding to a height of MUT between 3 cm and 4.5 cm). The resulting height was recorded to make sure that the electrodes were fully covered by the mixture. When the MUT is compressed, its volume changes in response to the applied pressure. As a result, the volumetric water and bentonite contents between the opposing electrodes offer more insightful information than the gravimetric contents, as discussed in our previous study [19]. To account for this, the previously mentioned water and bentonite mass contents were converted into their corresponding volumetric contents as follows:(1)$$\theta_{W}=\frac{m_{W}}{V}=\frac{m_{D}\cdot \omega_{W}}{V}=\rho_{D}\cdot \omega_{W},$$
(2)$$\theta_{BC}=\frac{m_{BC}}{V}=\frac{m_{D}\cdot \omega_{BC}}{V}=\rho_{D}\cdot \omega_{BC}$$
where $\theta_{W}$ and $\theta_{\mathrm{BC}}$ respectively are the volumetric water and bentonite contents, $\omega_{W}$ and $\omega_{\mathrm{BC}}$ are the corresponding mass contents, $m_{W}$, $m_{\mathrm{BC}}$, and $m_{D}$ are the masses of water, bentonite, and dry mixture in the measurement volume *V*, and $\rho_{D}$ is the density of dry mixture in the measuring cell.

Each sample was then characterized within the measuring cell before being removed. Impedance measurements were carried out on each sample using two repeated frequency sweeps. At the end, a total of 230 × 2 impedance spectra were available. The frequency sweep covered a range from 20 Hz to 1 MHz, with 201 discrete frequency points per sweep, and each sweep required approximately one minute to complete. At each frequency, the apparent impedance |*Z*| and the impedance phase φ were recorded. These values were then processed in Python 3.10.6 to compute the corresponding real (*R*) and imaginary (*X*) components of the impedance. Although lower frequencies (as low as 0.01 mHz) provide more detailed information on electrochemical systems [30,31,32], they require significantly longer measurement times. To ensure the transferability of our laboratory results to subsequent applications in the field, we prioritized rapid measurements to align with the brief intervals between foundry sand charges and the time variance of the sand properties during processing. Measurements were performed at room temperature (between 21 °C and 26 °C). A broader temperature range is not necessary because the temperature of the foundry material after mixing is usually between 20 °C and 40 °C. An impedimetric investigation of the MUTs in this temperature window did not reveal any significant influence of the temperature on the data [33].

## 3. Results and Discussion

### 3.1. Illustration of Impedance Measurements

For Figure 3, twelve samples were selected, representing three levels of bentonite content (2 g/100 mL, 6 g/100 mL, and 10 g/100 mL) and four levels of water content (2 g/100 mL, 4 g/100 mL, 6 g/100 mL, and 8 g/100 mL), all evaluated at the same dry density. The corresponding impedance *Z(f) = R +* j *X * (with X=−1/ωC), represented by its real component (resistance *R*) and imaginary component (reactance *X*), is depicted in the figure. The figure illustrates the measured frequency-dependent resistance *R* (Figure 3a) and reactance *X* (Figure 3b) as functions of bentonite and water contents. At water contents up to 5%, both the absolute value of *R* and *X* increase with higher bentonite content. This behavior can be attributed to the confined environment created within the montmorillonite interlayer spaces at low water levels. Such confinement restricts ion mobility, leading to a decrease in the diffusion coefficient as ion concentration increases [34]. Additionally, some sodium ions are absorbed directly into the clay surface, further reducing their mobility, contributing to an increase in resistance and resulting in lower conductivity. Additionally, at low water content the water molecules are bound to the surface of the mineral-like montmorillonite. This bound water layer reduces the availability of free water for ion movement, further increasing both *R* and *X* [35,36,37].

Vice versa, with the increase of water content, both the real (*R*) and imaginary (X=−1/ωC) components of the impedance decrease due to several factors. Higher water content leads to the expansion of the montmorillonite interlayer spaces, reducing the confinement of ions and enhancing their mobility [34], which decreases the resistance (*R*). Additionally, the dilution of ion concentration reduces ion–ion interactions, further lowering the resistive component. As the water content increases, ions in bentonite become more hydrated and dissociate more readily, promoting greater ionic conductivity. The availability of more water also reduces ion adsorption into the clay surfaces, allowing for greater ion mobility.

The data presented in Figure 3 indicate that the designed impedance sensor has the potential to measure bentonite content, possibly in field settings rather than solely in laboratory environments, provided proper calibration is performed. However, calibrations to account for variables such as specific foundry materials, water content, and applied pressure would be time-consuming and labor-intensive. To overcome this challenge, we explored methods to eliminate the need for extensive calibration. Our approach is based on the fact that impedance spectra contain a wealth of signal features. This richness in information makes it feasible to estimate both analyte contents and influence quantities (e.g., water content or density) simultaneously. Automating this estimation process necessitates the use of machine learning algorithms, FCNNs in particular. They are well suited for handling intricate, non-linear dependencies within the data, ensuring accurate and reliable estimation. This integration not only enhances precision but also minimizes the need for extensive human intervention, making it a powerful solution for bentonite content estimation.

A critical prerequisite for applying machine learning is data preparation, which includes reformatting the data, as well as performing data engineering tasks such as feature extraction and feature selection. Before the features were passed to the algorithm, they were preprocessed through centering and scaling to achieve a zero mean and unit standard deviation, ensuring consistency and improving model performance.

### 3.2. Feature Extraction

The raw data collected from our experiments consisted of measured impedance spectra *Z(f) = R +* j *X* (460 spectra from 230 foundry mixtures, each spectrum including 201 real and imaginary components, respectively), along with physical properties for each mixture (volumetric water content $\theta_{w}$, density $\rho_{b}$). Although it is possible to use raw EIS data directly as input for machine learning models such as classifiers [38], this leads to practical challenges. For effective model training, the dimensionality of the input data must not be too high, and there must be a sufficient number of observations to estimate the parameters reliably. High-dimensional datasets, such as raw EIS data containing 402 dimensionalities in our case, can often contain irrelevant features that act as noise, leading to poorer generalization in machine learning models [39]. This issue will be validated in our dataset, with results presented in Table 1.

Given the limited size of our EIS dataset, dimensionality reduction appeared essential. Two main strategies are commonly used for this purpose. One is feature extraction, which transforms raw data into a more effective set of inputs. It creates features from existing data by reducing data dimensionality through methods like Principal Components Analysis (PCA), Independent Component Analysis (ICA), and Linear Discriminant Analysis (LDA), and selecting the most relevant features for model training based on importance scores and correlation matrices [40,41,42]. Another approach is feature selection, which focuses on identifying the most relevant and important features from a given set while removing redundant or irrelevant ones. It is used to simplify the model, reduce overfitting, and improve computational efficiency. Common variants include filter methods (e.g., correlation-based selection), wrapper methods (e.g., recursive feature elimination), and embedded methods (e.g., feature importance in tree-based models) [42,43].

In our study, the input data included both raw EIS spectra and the additional physical properties $\theta_{w}$, density $\rho_{b}$ of each mixture. Since these physical properties significantly influence impedance measurements and, consequently, the estimation of bentonite content, we retained them as critical features. To further reduce the dimensionality of the raw EIS data, we employed PCA. PCA simplifies complex datasets by transforming them into a smaller set of new, uncorrelated variables called principal components, which retain most of the original information. By maximizing the variance in the data, PCA identifies the most significant patterns, solving an eigenvalue and eigenvector problem to derive components directly from the dataset rather than relying on predefined features [40]. This adaptability makes PCA an efficient and versatile tool for data analysis.

Figure 4 illustrates the workflow used in this study. First, the raw EIS data, represented by resistance *R* and reactance *X*, are normalized and processed using PCA to extract key features. From the raw EIS data comprising 402 features, the first ten principal components, which collectively explain 99.9% of the variance in the data, were extracted as a compact representation of the EIS data. These extracted components, along with the volumetric water content $\theta_{w}$, and the density $\rho_{b}$, are then used as input features to train FCNNs. Finally, permutation importance is applied to rank the features used for training, providing insights into the contributions of both the extracted EIS data and the physical properties to the model performance.

### 3.3. Training Regression Models Using Neural Networks

This section is devoted to evaluating the performance of various combinations of feature sets in estimating bentonite content in foundry mixtures. FCNNs were developed and implemented using the TensorFlow package (version 2.17.0) in Python 3.10.6. The models were trained on a desktop equipped with an Intel UHD 630 GPU.

The structure of the FCNNs is illustrated in Figure 4. It consists of three dense layers, with the ReLU activation function applied between each layer to introduce non-linearity and enhance the model’s learning capability. To train the FCNNs, the 460 observations were randomly split into training and testing datasets in a ratio of 8:2. To ensure the robustness and generality of the neural networks, a 10-fold cross-validation (*k* = 10) approach was used [44], particularly for handling potential outliers in the measurement data. This method helps prevent overfitting and ensures reliable performance across different feature set combinations.

As already discussed, the initial input features consisted of raw EIS data combined with volumetric water content $\theta_{w}$ and density $\rho_{b}$, resulting in a total of 404 features. To evaluate the regression performance of the FCNNs, the R-squared score (*R*^2^) was used as the evaluation metric, providing a measure of how well the model predictions match the actual values. A higher *R*^2^ value indicates a better predictive accuracy and regression performance of the network. Using the full data set, an average *R*^2^ score of 0.76 from 10-fold cross validation was achieved (Table 1). This indicates that the information contained in the data correlates with the bentonite content. However, the performance is far from satisfying. As previously discussed, a high feature dimensionality may contain noise and weaken the generalization of the networks.

To address this, new FCNNs were trained using the extracted 10 key features along with $\theta_{w}$ and $\rho_{b}$. This approach yielded significantly improved results, achieving an average *R*^2^ score as high as 0.94 (Table 1). This indicates that the FCNN trained by this reduced combination of features can accurately estimate the bentonite content in foundry materials.

**Table 1 sensors-24-08111-t001:** *R*^2^ of the bentonite content estimation with different combinations of feature set.

Feature Set	Number of Features	*R* ^2^
Raw EIS data + $\rho_{b}$ + $\theta_{w}$	404	0.76
Extracted EIS + $\rho_{b}$ + $\theta_{w}$	12	0.94
Extracted EIS + $\theta_{w}$	11	0.92
Extracted EIS + $\rho_{b}$	11	0.91
Extracted EIS	10	0.87

The actual and the predicted bentonite contents generated from the test dataset are illustrated in Figure 5.

### 3.4. Feature Selection for Estimating Bentonite Content

To gain insight into the contributions of EIS data details and physical properties to the model’s performance, feature importance was evaluated by the Leave-One-Feature-Out (LOFO) approach [45]. This method systematically assesses the impact of each feature by removing it and observing the change in model performance. In the LOFO evaluation, each feature is “removed” by setting its values to zero in a copy of the dataset. The model performance is then re-evaluated on this modified dataset, and the difference in the *R*^2^ score between the full feature set (${R^{2}}_{Full}$) and the reduced set (${R^{2}}_{LOFO}$) is calculated. This difference defines the importance score *S* as follows:(3)$$S={R^{2}}_{Full}-{R^{2}}_{LOFO}.$$It quantifies the importance of the removed feature, with larger drops indicating greater significance to the model.

To ensure reliability, the LOFO evaluation was repeated 15 times, which means the ${R^{2}}_{LOFO}$ is calculated 15 times. The average importance scores across all 15 iterations were computed. This repeated calculation reduces variability and provides a more robust estimation of the contribution of each feature to the model’s performance. The average importance scores are illustrated in Figure 6. The results clearly demonstrate that the four most important features for estimating bentonite content are the first three and the fifth principal components extracted from raw EIS data. These features have a significant impact on the performance of the FCNNs. Additionally, the physical properties—volumetric water content $\theta_{w}$ and density $\rho_{b}$—also play a crucial role in improving the estimation accuracy.

To validate the importance scores, FCNNs were trained using different feature combinations, as summarized in Table 1. Models trained with the extracted features combined with water content $\theta_{w}$ achieved an impressive average *R*^2^ score of 0.92, indicating that EIS data and water content alone can accurately estimate bentonite content without density information. Similarly, combining extracted features with density produced slightly weaker but still excellent results, with an average *R*^2^ score of 0.90.

Even when using only the extracted key features from the raw EIS data, the FCNNs achieved an average *R*^2^ score of 0.87. This aligns well with the calculated importance scores shown in Figure 6, further validating the contributions of the selected features to the model’s performance.

The LOFO evaluation highlights the significant contribution of the key features extracted from raw EIS data, as well as the physical properties (water content and density), in accurately estimating bentonite content. By identifying the most important features from complex datasets, this approach enhances both the efficiency and accuracy of the estimation process. This study demonstrates that high-quality estimations can be achieved using key features from raw EIS data combined with physical properties like water content and density. Even without one of the physical properties, accurate predictions of bentonite content (with *R*^2^ > 0.9) are still possible. This approach not only reduces measurement requirements but also simplifies the estimation process, making it more practical and cost-effective for industrial applications.

## 4. Conclusions

This study highlights the effectiveness of an AI-enhanced EIS approach in accurately estimating bentonite content in foundry materials, particularly when combined with physical properties such as water content and density. The regression FCNN model achieved a high *R*^2^ score of 0.94, demonstrating its suitability for meeting the requirements of foundry plants.

It was observed that directly using raw impedance data in machine learning models resulted in poor estimations, but through proper feature extraction and the inclusion of physical properties, accurate predictions of bentonite content were achieved. Remarkably, even without one of the physical properties, reliable predictions were still possible. This approach reduces the need for extensive measurements, simplifies the estimation process, and enhances its practicality and cost-effectiveness for industrial applications.

However, the study acknowledges that the controlled laboratory results cannot be directly applied to real-world foundry environments due to the high demands for quality data in EIS measurements. Factors such as noise, dust, and environmental interferences can significantly affect the accuracy of the regression model, despite its robustness in controlled settings. To address these challenges, we are adapting the measurement setup for in situ use in foundries, incorporating the Analog Discovery hardware from Digilent to conduct EIS measurements under field conditions. Alongside these measurements, water content and density will also be recorded for a comprehensive data set. The goal is to develop a reliable system capable of operating under the demanding conditions of a foundry while maintaining the accuracy observed in laboratory tests.

Future work will focus on assessing the generalizability of this approach across different foundry materials and various environmental conditions, further improving its applicability in industrial settings.

## Figures and Tables

**Figure 1 sensors-24-08111-f001:**
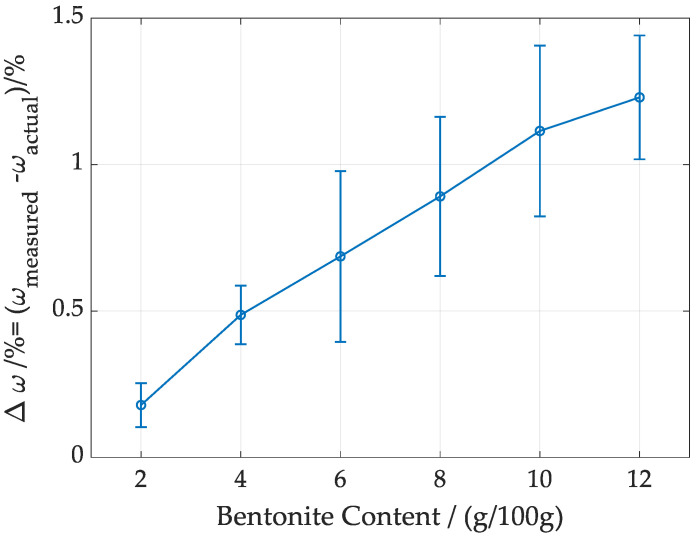
Average and standard uncertainty of the difference △ω between the water content $\omega_{measured}$ measured by oven-drying at 105 °C and the true water content $\omega_{actual}$ determined by weight measurements at room temperature.

**Figure 2 sensors-24-08111-f002:**
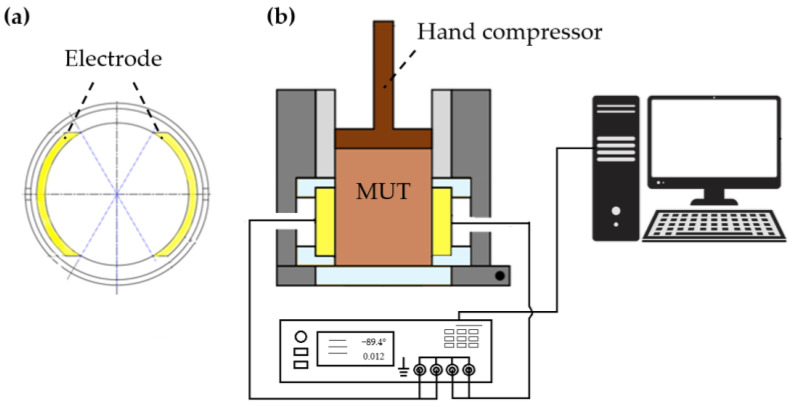
Impedance sensor. (**a**) Cross section of the opposite electrodes from the top view without MUT. (**b**) Measuring system consisting of a sensor, LCR meter, and computer.

**Figure 3 sensors-24-08111-f003:**
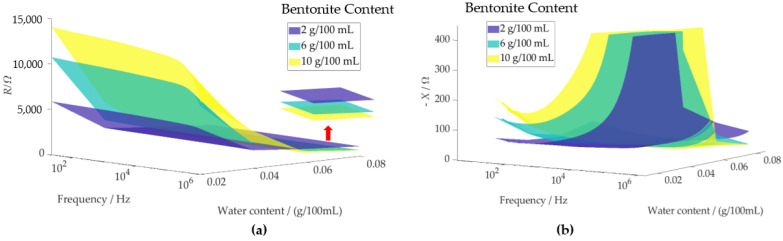
Measured test cell resistance (**a**) and reactance (**b**) as a function of bentonite content (in g/100 mL), water content (in g/100 mL), and frequency in foundry sands, evaluated at the same dry density for the twelve samples presented.

**Figure 4 sensors-24-08111-f004:**
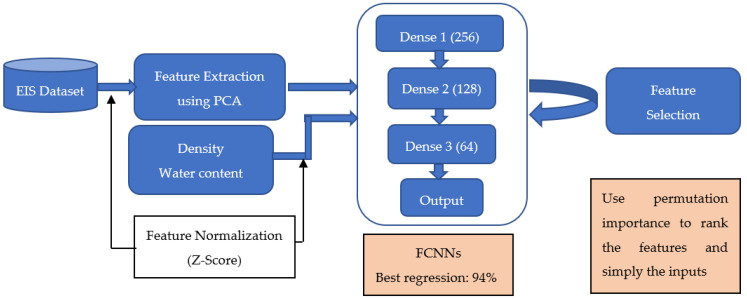
Overview of the workflow to estimate bentonite content using EIS data and selected physical properties.

**Figure 5 sensors-24-08111-f005:**
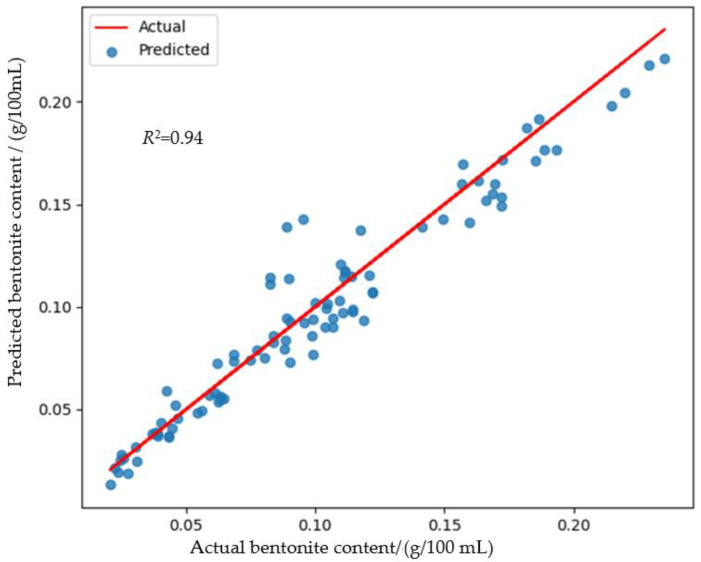
Predicted bentonite content using the developed FCNN regression model, based on key features extracted from impedance spectra, along with the water content and the density of synthetic molding material samples.

**Figure 6 sensors-24-08111-f006:**
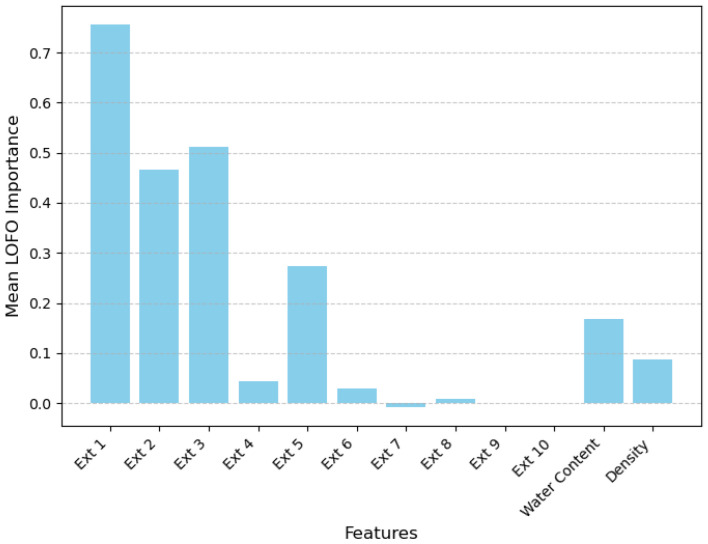
Average importance scores of each feature from the combination of extracted EIS data, water content, and density. Ext 1 through 10 designate the ten key features extracted from raw EIS data.

## Data Availability

The data are accessible online with the following DOI: https://doi.org/10.57880/rdspace-ubt-3, accessed on 19 November 2024.

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
