# Peer review of "Determination of the Bentonite Content in Molding Sands Using AI-Enhanced Electrical Impedance Spectroscopy"

_sensors, 2024, doi:10.3390/s24248111_

Round 1
Reviewer 1 Report
Comments and Suggestions for Authors
The manuscript entitled “Determination of the bentonite content in molding sands using AI-enhanced electrical impedance spectroscopy” reports the AI-enhanced electrical impedance spectroscopy for the determination of bentonite content in molding sand samples. While the manuscript has its merits, there are a few comments about certain aspects. The manuscript should be revised according to the following comments:
1) Error bars should be included in the data depicted in Fig.1.
2) In Fig. 3 justifies the differences in resistance and reactance properly. However, it would be more beneficial for the manuscript to express reactance (and in general imaginary impedance) in conjunction with capacitance, as per:
X=1/ωC
Provided that the system is based on a simple R-C in series behavior (since no faradaic reaction is involved), then this transformation can more easily be used as a function of ion concentration and diffusion.
Author Response
Thanks for your kindly review. According to your review, I did the following changes:
- Error bars should be included in the data depicted in 1
Thanks for the advice and we added an error bar for the measurement to provide more information, as shown in Figure 1.
- In 3 justifies the differences in resistance and reactance properly. However, it would be more beneficial for the manuscript to express reactance (and in general imaginary impedance) in conjunction with capacitance, as per:
X=1/ωC
The reactance X is expressed as X=-1/wC which is added in line 205.
Reviewer 2 Report
Comments and Suggestions for Authors
I found this manuscript highly relevant and closely related to one of my own studies. I enjoyed the work and would be interested in discussing it further with the authors if published. I recommend this paper for publication but suggest improvements in figures, methodology descriptions, and result analysis to enhance its clarity and impact.
1. Please ensure consistency in all figures, such as including boundaries for axes (top, bottom, left, and right) and maintaining uniform axis label formatting. Most figures seem to be generated using Python's Matplotlib. I suggest exporting the data and using professional scientific plotting software, such as Origin, to achieve better consistency and visual quality.
2. I recommend adding clear indications of the electrode and sample positions in Figure 2a to enhance the figure's clarity and provide better context for the experimental setup.
3. Consider including representative impedance spectrum plots or Nyquist plots (EIS) for selected samples. This would help readers better understand the experimental results and data interpretation.
4. The axis labels in Figure 3 are somewhat unconventional. Typically, the units should be enclosed in parentheses next to the corresponding axis information.
5. Figure 3 is not sufficiently informative. It is unclear how the signal within the specified frequency range relates to the water content. For instance, different colors appear to represent water content, yet one axis also represents water content, which is confusing. I suggest revising the figure to present the relationship more clearly. This figure should be used convincing the readers that the EIS spectrum is sensitive to the water content.
6. Before introducing the model, the authors could consider using illustrative diagrams or legends to explain the features, such as the real and imaginary components and how they are combined. Maybe can include in the figure 4.
7. PCA is a powerful and effective method, and I suggest the authors visualize the PCA results to provide more insights—this could reveal interesting patterns or trends. Additionally, clarify whether the extracted features correspond to different PCA components. A detailed explanation on this point would help readers better understand the feature selection process.
8. Please provide more details on the data distribution during database construction. For example, how is the water content distributed, and are there any other differences among the data samples beyond water content? To enhance clarity, consider marking these distinctions in Figure 5. For instance, use scatter plot colors to indicate different categories or attributes. This could help identify whether certain types of samples are more challenging to predict.
9. In the current era of AI, it is crucial to justify the use of complex models. Why it is necessary to use AI algorithm and why do you think it is helpful? Including comparisons with baseline algorithms would strengthen the manuscript.
10. I want to emphasize again the need for improving the quality of the figures. The figures need to be more informative and intuitively designed to convey key messages effectively. Improving figure quality will enhance the paper’s clarity. The mechanistic explanation is somewhat shallow; consider addressing this as a future research direction.
Comments on the Quality of English Language
The introduction part can be more starightfoward. Help to reader get what kind of challenge you want to address.
Author Response
Thanks for your review. Please see the attachment.
We are glad to discuss it after the publishment.

Reviewer 3 Report
Comments and Suggestions for Authors
The manuscript “Determination of the bentonite content in molding sands using AI-enhanced electrical impedance spectroscopy” developed a method to determine the bentonite concentration mixed with quartz sand and deionized water under laboratory conditions, by using AI to analyze both the real and imaginary components of the impedance measured at different frequency range from 20 Hz to 1 MHz. The authors applied principal component analysis to extract 10 key features and used the 10 features plus the water content and density as input data to train machine-learning models, leading to a high prediction accuracy at 0.94. The manuscript is clearly written, I would recommend its publication on Sensors with the following minor revisions.
1. In line 119, add the product numbers for sodium bentonite and quartz sand.
2. In line 121, what is the resealable plastic bag used in experiments?
3. What is the volume and weight range of the 230 samples?
4. In line 192, please indicate the 4 water content levels for Figure 3.
5. In line 308, please add a reference for the 10-fold cross-validation (k=10) approach.
6. In line 339, please add a reference for the LOFO approach.
7. In line 347, please explain how the average importance scores in Figure 6 are calculated.
